# Constructing a T-Cell Receptor-Related Gene Signature for Prognostic Stratification and Therapeutic Guidance in Head and Neck Squamous Cell Carcinoma

**DOI:** 10.3390/cancers15235495

**Published:** 2023-11-21

**Authors:** Ye Lu, Zizhao Mai, Jiarong Zheng, Pei Lin, Yunfan Lin, Li Cui, Xinyuan Zhao

**Affiliations:** 1Stomatological Hospital, School of Stomatology, Southern Medical University, Guangzhou 510280, China; 3170040041@smu.edu.cn (Y.L.); maizizhao@smu.edu.cn (Z.M.); linpei@smu.edu.cn (P.L.); linyunfan@smu.edu.cn (Y.L.); 2Department of Dentistry, The First Affiliated Hospital, Sun Yat-Sen University, Guangzhou 510080, China; zhengjr26@mail2.sysu.edu.cn

**Keywords:** T-cell receptor-related genes, head and neck squamous cell carcinoma, risk signature, prognosis prediction, immune infiltration

## Abstract

**Simple Summary:**

The accurate stratification of head and neck squamous cell carcinoma (HNSCC) patients based on prognostic differences, using robust biomarkers or signatures, is crucial for guiding clinical interventions. Our study aimed to develop a predictive signature for head and neck squamous cell carcinoma outcomes based on T-cell receptor-related genes (TCRRGs). Using The Cancer Genome Atlas HNSCC dataset, GSE41613, and GSE65858, we identified two HNSCC clusters based on TCRRG expression. These clusters showed differences in overall survival (OS) and immune infiltration. A robust TCRRG-based prognostic signature comprising MAP2K7, MAPK3, MAPK9, ORAI1, PSMA1, UBB, and ZAP70 was subsequently constructed and validated across multiple HNSCC cohorts. A nomogram model was then constructed for personalized HNSCC treatment guidance. Functional analyses indicated notable changes in biological functions and pathways between high- and low-risk groups, with the high-risk group exhibiting a suppressive immune environment. Utilizing this TCRRG-based signature, we may precisely forecast HNSCC outcomes, offering enhanced therapeutic strategies.

**Abstract:**

Backgroud: The stratification of head and neck squamous cell carcinoma (HNSCC) patients based on prognostic differences is critical for therapeutic guidance. This study was designed to construct a predictive signature derived from T-cell receptor-related genes (TCRRGs) to forecast the clinical outcomes in HNSCC. Methods: We sourced gene expression profiles from The Cancer Genome Atlas (TCGA) HNSCC dataset, GSE41613, and GSE65858 datasets. Utilizing consensus clustering analysis, we identified two distinct HNSCC clusters according to TCRRG expression. A TCRRG-based signature was subsequently developed and validated across diverse independent HNSCC cohorts. Moreover, we established a nomogram model based on TCRRGs. We further explored differences in immune landscapes between high- and low-risk groups. Results: The TCGA HNSCC dataset was stratified into two clusters, displaying marked variations in both overall survival (OS) and immune cell infiltration. Furthermore, we developed a robust prognostic signature based on TCRRG utilizing the TCGA HNSCC train cohort, and its prognostic efficacy was validated in the TCGA HNSCC test cohort, GSE41613, and GSE65858. Importantly, the high-risk group was characterized by a suppressive immune microenvironment, in contrast to the low-risk group. Our study successfully developed a robust TCRRG-based signature that accurately predicts clinical outcomes in HNSCC, offering valuable strategies for improved treatments.

## 1. Introduction

Head and neck squamous cell carcinoma (HNSCC) is a pervasive and aggressive cancer type originating from the epithelial layers of the oral cavity, larynx, and pharynx [1]. The occurrence of HNSCC is predominantly associated with risk factors such as genetic susceptibility, infection with human papillomavirus, tobacco smoking, betel nut chewing, and heavy alcohol consumption [2,3]. At present, HNSCC treatment strategies encompass surgical intervention, chemotherapy, radiotherapy, targeted therapy, and immunotherapy [4]. Nonetheless, despite a broad array of therapeutic tools, the 5-year survival rate of HNSCC patients has demonstrated minimal progress over time [5]. Therefore, elucidating the molecular mechanisms driving HNSCC malignancy and establishing novel prognostic biomarkers of improved precision is crucial [6,7,8,9]. The development of sophisticated biomarker profiles is fundamental for enhancing clinical outcomes for HNSCC patients, as it enables the selection of personalized treatments, minimizes the risk of adverse reactions, and reduces unnecessary healthcare costs.

Tumor immunotherapy has emerged as a promising therapeutic approach for patients with recurrent or metastatic HNSCC [10]. Immune checkpoint inhibitors (ICIs) have significantly advanced tumor immunotherapy by interrupting signals that allow cancer cells to evade immune detection [11]. The most extensively researched ICI therapies target the programmed cell death 1 receptor (PD-1) and its ligand (PD-L1), which act as a braking mechanism for the immune system [12]. This approach has been successful in treating various cancers, including HNSCC, lung cancer, liver cancer, and breast cancer [13,14,15]. Specifically, PD-1/PD-L1 inhibitors have shown significant survival benefits in HNSCC, enhancing both overall and progression-free survival rates [16]. Employed either as monotherapy or in combination with other treatments, these therapies have become integral to clinical practice, demonstrating considerable potential in cancer management [17]. The effectiveness of cancer immunotherapy largely rests on the crucial role of T-cells, which are essential for identifying and responding to tumor-associated antigens [18]. T-cell receptors (TCRs), an extensive assortment of antigen receptors on T-cell surfaces, are integral for immune response as they recognize and bind to specific antigens presented by major histocompatibility complex molecules on antigen-presenting cells [19]. The exceptional diversity of TCRs originates from a process called V(D)J recombination, which generates a vast repertoire of TCRs, each with unique antigen-binding capabilities [20]. Thus, precise and effective TCR targeting of tumor antigens presents a potent therapeutic strategy to enhance clinical outcomes for HNSCC patients [21]. TCR-related genes (TCRRGs) are those genes directly or indirectly associated with TCR [22]. For example, cytotoxic T lymphocyte antigen-4 (CTLA-4) is an immune checkpoint molecule predominantly expressed on T-cells, and CTLA-4 blockade combined with radiotherapy significantly modulates the TCR repertoire of tumor-infiltrating T-cells [23]. Similarly, T-cell activation via TCRs upon antigen recognition leads to an increase in intracellular calcium concentration, and this process is mediated by STIM1 and Orai1 [24].

In the present study, we established a TCRRG-based risk prediction model employing The Cancer Genome Atlas (TCGA) HNSCC train cohort, with its robustness successfully corroborated in the TCGA HNSCC test cohort, GSE41613, and GSE65858. Additionally, a predictive nomogram model using the TCRRG risk signature was developed to accurately forecast the OS of HNSCC patients. Notably, our study unveiled a substantial rise in immune cell infiltration and improved immune functions within the low-risk group in contrast to the high-risk group. These observations shed light on the potential mechanisms contributing to the favorable clinical outcomes identified in the low-risk group.

## 2. Materials and Methods

### 2.1. Public Data Source

RNA-seq transcriptome data and associated clinicopathological information for HNSCC patients were acquired from the TCGA database. The clinical samples originated from diverse head and neck regions, including the oral cavity, tongue, floor of the mouth, tonsil, gum, palate, lip, larynx, oropharynx, pharynx, and hypopharynx. The study included a total of 497 HNSCC cases. Additionally, microarray data and relevant clinical information were obtained from the Gene Expression Omnibus databases using the accession numbers GSE41613 and GSE65858. The microarray data for GSE41613 (*n* = 97) and GSE65858 (*n* = 270) were based on the GPL570 (Affymetrix Human Genome U133 Plus 2.0 Array) and the GPL10558 (Illumina HumanHT-12 V4.0 expression beadchip, Illumina Inc., San Diego, CA, USA), respectively. All patients in the TCGA HNSCC cohort underwent surgery, and a subset of these patients also received postoperative radiotherapy. Within the GSE41613 cohort, 43 patients were treated with a multi-modality, whereas 53 patients underwent a combination of treatment modalities. In the GSE65858 cohort, 78 patients received a single therapeutic approach, 189 patients were treated with multiple modalities, and 3 patients received palliative care, though specific treatment details were not provided. The TCR-related genes were obtained from the ImmPort Portal (https://www.immport.org/home, accessed on 8 December 2022) and GSEA databases (GSEA software version 4.3.2), including REACTOME_TCR_SIGNALING, BIOCARTA_TCR_PATHWAY, PID_TCR_PATHWAY, and KEGG_T_CELL_RECEPTOR_SIGNALING_PATHAWY.

### 2.2. TCRRG Risk Signature: Construction and Validation

The TCGA HNSCC cohort was randomly split into a train cohort and a test cohort. Univariate Cox analysis was employed to examine the correlation between TCRRGs and OS in the TCGA HNSCC train cohort. The most optimal TCRRGs were selected using the least absolute shrinkage and selection operator (LASSO) regression method. A unique TCRRG-based prognostic signature was derived via multivariate Cox proportional hazards regression analysis. A risk score for each patient was computed as the sum of each gene’s score, derived by multiplying the expression of each gene by its coefficient. The TCGA HNSCC train cohort was divided into high-risk and low-risk groups based on the median value of risk scores. The Kaplan–Meier method and log-rank test were utilized to calculate the difference in OS between these groups. Moreover, a comparison of OS, stratified by various clinicopathological parameters between these groups, was conducted. A receiver operating characteristic (ROC) curve was generated to assess the prediction accuracy of the prognostic model. The utility of the risk score as an independent prognostic factor was analyzed using univariate and multivariate Cox regression analyses. Finally, the robustness of the TCRRG-based prognostic signature was validated in independent HNSCC cohorts, which included the HNSCC test cohort, GSE41613, and GSE65858.

### 2.3. Nomogram Model Construction

A nomogram model, incorporating risk scores and clinicopathological parameters such as age, gender, tumor grade, and TNM stage, was established. Calibration curves were employed to assess the accuracy of predicting the 1-year, 2-year, and 3-year OS rates in HNSCC.

### 2.4. Pathway and Function Enrichment Analysis

A volcano plot was utilized to depict differentially expressed genes (DEGs) between high- and low-risk groups. Following this, Gene Ontology (GO) and Kyoto Encyclopedia of Genes and Genomes (KEGG) analyses were conducted to pinpoint enriched biological processes and pathways. These analyses were carried out using the “clusterProfiler” R package (R software version 4.2.3).

### 2.5. Immune Cell Infiltration Analysis

In order to calculate the stromal score, immune score, and tumor purity in high-risk and low-risk groups, the ESTIMATE R package (R software version 4.2.3) was employed. The ggplot2 R package (R software version 4.2.3) was used to illustrate these differences. The CIBERSORT method was performed to determine the disparities in infiltration abundance of 22 immune cell types within high-risk and low-risk groups. Violin plots, derived from the ggplot2 R package (R software version 4.2.3), graphically displayed these variations in immune infiltrating cell types among distinct subgroups. Furthermore, the immune function in the low-risk and high-risk groups was estimated via single sample Gene Set Enrichment Analysis (ssGSEA). The most pronounced positive and negative correlations between risk scores and immune cell types were visualized with a scatter plot.

### 2.6. Mutation Landscape of High- and Low-Risk Groups

The mutation profiles of HNSCC samples were sourced from the TCGA Data Portal, and the ‘maftools’ package was used to contrast the mutational profiles between high-risk and low-risk groups.

### 2.7. Prediction of Immunotherapy Efficacy

The Tumor Immune Dysfunction and Exclusion (TIDE) web platform (http://tide.dfci.harvard.edu, accessed on 8 December 2022) was utilized to evaluate the potential immune therapy response in the TCGA HNSCC cohort. By following procedures provided by the TIDE platform, median centering of the expression matrix for both low-risk and high-risk groups was performed and submitted for analysis. This yielded important metrics such as cancer-associated fibroblast (CAF) score, T-cell exclusion score, and T-cell dysfunction score, all of which were compared between the two risk groups. 

### 2.8. Statistical Analyses

R software (version 4.2.3) was used for all statistical analyses. To evaluate the difference in OS between specified groups, we applied Kaplan–Meier curves and performed the log-rank test. Additionally, we utilized the two-sided t-test to compare the differences between the two groups. A *p*-value < 0.05 was considered statistically significant.

## 3. Results

### 3.1. Construction and Validation of TCRRG-Based Risk Signature

Univariate Cox regression analysis revealed a significant correlation between the OS of HNSCC patients and several TCRRGs, namely *CSF2*, *INPP5D*, *MAP2K1*, *MAP2K7*, *MAPK3*, *MAPK9*, *ORAI1*, *PIK3R3*, *PSMA1*, *PSMA7*, *PSMD10*, *PSMD2*, *PSMD7*, *SKP1*, *UBB*, *UBE2D2*, and *ZAP70* (Table 1). Among these, *CSF2*, *MAP2K1*, *MAPK9*, *ORAI1*, *PSMA1*, *PSMA7*, *PSMD10*, *PSMD2*, *PSMD7*, *SKP1*, *UBB*, and *UBE2D2* were identified as risky genes (HR > 1). In contrast, *INPP5D*, *MAP2K7*, *MAPK3*, *ORAI1*, *PIK3R3*, and *ZAP70* were deemed as protective genes (HR < 1). For the construction of the prognostic signature, seven TCRRGs—*MAP2K7*, *MAPK3*, *MAPK9*, *ORAI1*, *PSMA1*, *UBB*, and *ZAP70*—were selected, and their coefficients were obtained from the LASSO algorithm (Table 2, Appendix A). The risk score for each patient was calculated using the following formula: risk score = (−0.72216) × *MAP2K7* expression + (−0.484) × *MAPK3* expression + (0.58118) × *MAPK9* expression + (−0.38196) × *ORAI1* expression + (0.477007) × *PSMA1* expression + (0.48365) × *UBB* expression + (−0.46462) × *ZAP70* expression. On the basis of the median risk score value, HNSCC patients were divided into high-risk and low-risk groups following the computation of risk scores (Appendix A). To visually represent the survival time and status of every HNSCC patient in the TCGA HNSCC train cohort, a scatter plot was depicted (Figure 1A). A heatmap was then used to present the differential expression levels of the seven prognostic TCRRGs between low-risk and high-risk groups within the TCGA HNSCC train cohort (Figure 1B). The results of the survival analysis revealed that patients defined in the high-risk group had a significantly shorter OS (*p* = 1.353 × 10^−6^) compared to those in the low-risk group (Figure 1C). This TCRRG-based prognostic signature’s robustness was subsequently validated across several independent HNSCC cohorts, including the TCGA HNSCC test cohort (*p* = 1.621 × 10^−4^) (Figure 1D–F, Appendix A), GSE65858 (*p* = 4.029 × 10^−2^) (Figure 1G–I, Appendix A), and GSE41613 (*p* = 1.712 × 10^−3^) (Figure 1J–L, Appendix A). Following this, we delved further into the survival differences between low- and high-risk groups, performing stratification based on several clinicopathological parameters—age, gender, clinical stage, T stage, and N stage. In the TCGA HNSCC train cohort, the high-risk group exhibited a significantly lower OS rate than the low-risk group for patients aged ≤60 (*p* = 1.5 × 10^−2^) or >60 (*p* = 8.3 × 10^−3^), for male patients (*p* = 1.3 × 10^−3^), for patients at stages III–IV (*p* = 3 × 10^−4^) or T3–T4 (*p* = 1.1 × 10^−4^), and for patients with (*p* = 4.3 × 10^−2^) or without lymph node metastasis (LNM) (*p* = 5.3 × 10^−3^). Conversely, female patients (*p* = 7.6 × 10^−2^), those at stages I–II (*p* = 1.1 × 10^−1^), or at T1–T2 (*p* = 1.6 × 10^−1^) did not demonstrate significant differences in OS between high- and low-risk groups (Appendix A). For the TCGA HNSCC test cohort, the high-risk group showed significantly lower OS rates for patients aged >60 (*p* = 6.5 × 10^−4^), both male (*p* = 3.2 × 10^−2^) and female patients (*p* = 1.4 × 10^−2^), those at stages III–IV (*p* = 8.4 × 10^−4^) or T3–T4 (*p* = 4.7 × 10^−3^), and those with LNM (*p* = 2.8 × 10^−3^). However, no significant difference in OS was observed for patients aged ≤60 (*p* = 2.6 × 10^−1^), those at stages I–II (*p* = 4.6 × 10^−1^) or T1–2 (*p* = 1.4 × 10^−1^), and those without LNM (*p* = 1.5 × 10^−1^) (Appendix A). In the GSE65858 cohort, the high-risk group had significantly lower survival rates for male patients (*p* = 4.3 × 10^−2^), those at stages I–II (*p* = 1 × 10^−2^) or T1–T2 (*p* = 3.1 × 10^−2^), and those without LNM (*p* = 2.6 × 10^−2^). No significant difference in OS was identified for patients aged ≤60 (*p* = 1.1 × 10^−1^) or >60 (*p* = 1.8 × 10^−1^), female patients (*p* = 6.6 × 10^−1^), those at stages III–IV (*p* = 1.9 × 10^−1^) or T3-4 (*p* = 2.8 × 10^−1^), and those with LNM (*p* = 3 × 10^−1^) (Appendix A). In the GSE41613 cohort, significantly lower survival rates were observed in the high-risk group for patients aged ≤60 (*p* = 3.8 × 10^−2^) or >60 (*p* = 1.7 × 10^−2^), male patients (*p* = 2.8 × 10^−4^), and those at stage III–IV (*p* = 2.4 × 10^−4^). There was no significant difference in OS for female patients (*p* = 5.6 × 10^−1^) or those at stage I–II (*p* = 8.2 × 10^−1^) between the high- and low-risk groups (Appendix A). Moreover, we further explored the differences in OS between the low- and high-risk groups based on HPV status. In the TCGA HNSCC cohort and GSE65858, it was proven that the high-risk group exhibited a lower OS rate compared to the low-risk group among HPV-negative patients (*p* < 1 × 10^−3^). In contrast, for HPV-positive patients, there was no significant difference in OS between the high- and low-risk groups (Appendix A). We also conducted survival analyses based on various primary tumor sites in the TCGA HNSCC cohort. The results show a significant difference in OS between the high-risk and low-risk groups for HNSCC originating in the larynx and oral cavity (*p* < 1 × 10^−3^) (Appendix A).

### 3.2. The Predictive Accuracy of TCRRG-Based Risk Signature

To further scrutinize the predictive accuracy of the TCRRG-based risk signature, ROC curves were generated for OS prediction, and the corresponding areas under the curve (AUCs) were computed. Additionally, the predictive effectiveness of the risk signature was compared against various clinicopathological parameters, such as age, gender, grade, and stage. In the TCGA HNSCC train cohort, as demonstrated in Figure 2A, the risk score displayed substantial predictive capacity, reflected by 1-year, 3-year, and 5-year AUCs of 0.707, 0.705, and 0.685, respectively. Furthermore, the risk score outperformed other factors like age, gender, grade, and stage in terms of predictive effectiveness (Figure 2B). Similar observations were identified in the remaining validation cohorts, including the TCGA HNSCC test cohort (Figure 2C,D), GSE65858 (Figure 2E,F), and GSE41613 (Figure 2G,H), underscoring the robustness and effectiveness of the risk signature.

### 3.3. TCRRG-Based Risk Signature Is an Independent Prognostic Factor for HNSCC

We then evaluated whether the TCRRG-based risk signature was an independent prognostic factor for HNSCC cohorts. In the TCGA HNSCC train cohort, univariate analysis unveiled a significant association between the risk score (*p* < 1 × 10^−3^, HR = 1.504, 95% CI = 1.345–1.682) and OS (Figure 3A). The multivariate analysis further corroborated the risk score (*p* < 1 × 10^−3^, HR = 1.470, 95% CI = 1.309–1.651) as an independent prognostic indicator (Figure 3B). 

In the TCGA HNSCC test cohort, the univariate analysis demonstrated that both age (*p* = 3 × 10^−3^, HR = 1.028, 95% CI = 1.010–1.046) and risk score (*p* < 1 × 10^−3^, HR = 1.229, 95% CI = 1.121–1.346) were significantly correlated with survival (Figure 3C). Subsequent multivariate analysis confirmed age (*p* = 2 × 10^−3^, HR = 1.030, 95% CI = 1.011–1.049) and risk score (*p* < 1 × 10^−3^, HR = 1.256, 95% CI = 1.141–1.381) as independent prognostic determinants for HNSCC (Figure 3D). Likewise, in GSE65858 and GSE41613 cohorts, the risk scores were found to be significantly associated with survival (*p* = 4.1 × 10^−2^, HR = 1.540, 95% CI = 1.017–2.332; *p* = 0.002, HR = 2.423, 95% CI = 1.370–4.283, respectively) and served as independent prognostic predictors (*p* = 3.9 × 10^−2^, HR = 1.549, 95% CI = 1.023–2.344; *p* = 1 × 10^−3^, HR = 2.632, 95% CI = 1.459–4.747, respectively) for HNSCC patients (Figure 3E–H).

### 3.4. Nomogram Model Construction and Prediction

We further established a nomogram model incorporating risk score, age, gender, grade, and clinical stage to predict the clinical outcome of HNSCC. This model generates a total score for each HNSCC patient, calculated from their individual clinicopathological parameters and corresponding points. Consequently, the nomogram model facilitates the prediction of 1-year, 2-year, and 3-year OS rates for HNSCC patients (Figure 4A). The calibration curves demonstrate the good performance of our nomogram model, exhibiting excellent conformity in predicting 1-year, 2-year, and 3-year OS rates for HNSCC patients (Figure 4B–D).

### 3.5. Genomic Alterations in Low- and High-Risk HNSCC Groups and Their Impact on Survival Rates

At the genomic level, we found that 83.4% of HNSCC cases (196/235) had somatic mutations in the low-risk group. The most frequent alterations were *TTN*, *TP53,* and *CSMD3*. In the high-risk group, 87.06% of (222/255) HNSCC cases had somatic mutations, with *TTN*, *TP53*, and *MUC16* showing the highest frequency of alterations (Appendix A). Moreover, we observed that patients with both a lower TMB and risk score had improved OS rates compared to those with both a higher TMB and risk score in both the TCGA HNSCC train and test cohorts (Appendix A).

### 3.6. Functional Enrichment Analysis of the DEGs between High- and Low-Risk Groups

We then identified the DEGs between high- and low-risk groups in the TCGA HNSCC cohort, using a threshold of |log2FC| > 1 and FDR < 0.05 (Figure 5A). Our GO and KEGG analyses demonstrated that DEGs were significantly enriched in biological processes and signaling pathways related to immune activation and response. These biological processes and signaling pathways included immune response-activating cell surface receptor signaling, immune response-activating signal transduction, immune response-regulating cell surface receptor signaling, leukocyte-mediated immunity, and lymphocyte-mediated immunity. This underscores the crucial role of TCRRG alterations in shaping the tumor microenvironment of HNSCC (Figure 5B–E).

### 3.7. Correlation of Risk Score with Immune Cell Subpopulations and Potential for Immunotherapy Responsiveness

Recognizing the significant influence of tumor-infiltrating immune cells on the tumor immune microenvironment and their consequential impact on tumor progression, the correlation between the risk score and infiltrating immune cell subpopulations, as derived from the TCGA HNSCC database, was initially evaluated using the CIBERSORT algorithm. This examination revealed different levels of immune cell populations in the low-risk and high-risk groups. The low-risk group was characterized by a higher proportion of naïve B cells, plasma cells, CD8 T^+^ cells, activated memory CD4^+^ T-cells, and activated dendritic cells. In contrast, the high-risk group showed a higher proportion of resting memory CD4^+^ T-cells, M0 macrophages, and resting dendritic cells (Figure 6A,B). 

Additionally, the relative levels of CD8 T^+^ cells, T-cells, B lineage cells, NK cells, and cytotoxic lymphocytes were notably elevated in the low-risk group (Figure 6C). Considering the profound impact of immune cells on immune function, further comparison of ssGSEA scores for immune function revealed significantly higher immune function scores in the low-risk group compared to the high-risk group (Figure 6D). Using the ESTIMATE algorithms for an in-depth examination of the tumor immune microenvironment, it was discovered that the low-risk group displayed higher immune scores, reduced levels of T-cell dysfunction and exclusion, and a lower abundance of CAFs. This points towards a greater potential responsiveness to immunotherapy among HNSCC patients in the low-risk group (Figure 6E). Finally, a Pearson correlation analysis showcased positive associations between risk scores and certain immune cell types, namely resting memory CD4^+^ T-cells, M0 macrophages, eosinophils, resting dendritic cells, and resting mast cells. Conversely, negative associations were detected with other immune cell types, including active memory CD4^+^ T-cells, follicular helper T-cells, CD8^+^ T-cells, naïve B cells, plasma cells, and activated mast cells (Appendix A).

### 3.8. Correlation of Risk Score with T-cell Functions

The comparative analysis of T-cell function-related genes and chemokine expression profiles between the high-risk and low-risk groups was conducted. The results show a significant reduction in both T-cell function-related genes and chemokine expression within the high-risk group, underscoring the intimate correlation between TCRRG and the activities of T-cell cytotoxicity and recruitment (Figure 7A). These consistent results were also validated in the independent dataset GSE61413 (Figure 7B).

The GSEA results prominently illuminate that in the high-risk group, the enriched signatures associated with T-cell-related signaling pathways, including antigen receptor, B cell receptor, CD4+/CD8+ αβ T-cell lineage, immunoglobulin production, positive regulation of activated T-cell proliferation, positive T-cell selection, regulation of Th17 cell differentiation, T-cell lineage, T-cell receptor, and T-cell selection (Figure 7C). Consistently, the T-cell-related pathways encompassing adaptive immune response, antigen processing and presentation, B cell-mediated immunity, cell killing, leukocyte meditated cytotoxicity, leukocyte-mediated immunity, natural killer cell-mediated immunity, positive T-cell selection, regulation of natural killer cell-mediated immunity, and T-cell activation were markedly enriched in the high-risk group within the GSE41613 cohort (Figure 7D).

## 4. Discussion

The development of our TCRRG-based risk signature offers several distinct advantages. To the best of our knowledge, there currently are no existing TCR-based prognostic signatures available for predicting the prognosis of HNSCC. Thus, our unique signature provides a pioneering TCR-based solution. While many prognostic signatures incorporate a substantial quantity of genes, which can present obstacles to their clinical application, our risk signature only consists of seven genes. This reduction enhances the feasibility and utility of our signature for clinical application. Furthermore, we have not only developed this risk signature but have also successfully devised a TCR-based nomogram model. This model holds considerable potential for offering therapeutic guidance specially tailored to HNSCC patients. Lastly, our TCR-based prognostic signature, as shown through multivariate analysis, demonstrates superior robustness in predicting HNSCC prognosis in comparison to the traditional TNM stage classification, underscoring the clinical value of our signature as a reliable prognostic tool for HNSCC patients.

Recent research has increasingly shown the importance of the tumor immune microenvironment in cancer progression. Consequently, we investigated the potential correlation between our TCR-related prognostic signature and the immune tumor microenvironment. The GO and KEGG analyses of the DEGs between the high-risk and low-risk groups highlighted an enrichment of numerous immune-related biological processes and pathways. GSEA analysis between the high-risk and low-risk groups has revealed a significant enrichment of genes in the TCR-mediated pathway. Moreover, an immune-suppressive environment was apparent in the high-risk group, leading us to hypothesize that variances in the immune microenvironment among the patient groups contributed to their differential survival outcomes.

Through the analysis of TCRRGs affecting the OS in HNSCC, we identified MAPK9, PSMA1, and UBB as risk genes. MAPK9 is an important downstream signaling molecule in the TCR pathway and is involved in the activation and proliferation of T-cells. Silencing MAPK9 leads to impaired T-cell activation and insufficient secretion of cytokines. However, it has little effect on the apoptosis of mature T-cells [25]. MAP2K7 is a member of the serine/threonine protein kinase family, which can specifically phosphorylate and activate the MAPK family protein JNK [26]. Then, the activated JNK can participate in the positive transmission of TCR signals and promote the proliferation and differentiation of T-cells [27]. 

MAPK3 integrates cytoplasmic signals to induce transcriptional changes in the context of differentiation, proliferation, and survival [28]. It has been demonstrated that MAPK3 plays a key role in positive selection during T-cell development, and activation of MAPK3 is necessary for the induction of T-cell anergy [29,30]. Members of the MAP kinase family also play an important role in the innate and adaptive immune response by modulating the JNK pathway, which regulates T-cell differentiation and survival, mediates the response to viral and bacterial infections, and destroys cancer cells [31]. PSMA1 is one of the key regulatory subunits of the cytochrome C-terminal enzyme pathway [32]. During TCR signal transduction, PSMA1 expression is upregulated and promotes the degradation of downstream effector proteins in the TCR signaling pathway, thereby participating in TCR signaling-induced cell apoptosis. The ubiquitin B protein encoded by the UBB gene can participate in the conduction, regulation, and termination of TCR signaling by providing ubiquitination modification [33]. In the TCR signaling transduction process, ubiquitin B can ubiquitinate and precisely regulate the activity of some key effector molecules such as PKCθ, Bcl10, and MEKK1, thereby affecting TCR signal transduction. ORAI1 is one of the pore components of Ca^2+^-release-activated Ca^2+^ channels that mediate Ca^2+^ entry induced by TCR stimulation [34]. Following the depletion of intracellular calcium stores, T lymphocytes require a sustained influx of calcium ions via calcium release-activated calcium channels to develop an effective immune response [35]. At this point, Orai1 and STIM-1 aggregate and interact in close proximity to the plasma membrane, which can form the basic unit of Ca^2+^ entry and facilitate the formation of stable immune synapses between T-cells and antigen-presenting cells (APCs) [36,37]. Consequently, a number of T-cell surface and signaling molecules are recruited to the IS at the interface of APC, thereby optimizing the conditions under which they interact to amplify and maintain the signals required for full T-cell activation and efficiently generate an immune response [38,39]. Moreover, Lioudyno et al. showed increased Orai1 and STIM1 mRNA expression in activated T lymphocytes, which may contribute to enhanced signaling in activated T-cells [40].

The cytoplasmic tyrosine kinase ZAP70 is one of the additional signaling molecules that the TCR relies on to transduce signals. It can associate with the TCR complex and is required to initiate the canonical biochemical signal pathways downstream of the TCR [41]. Novotná et al. identified the ability of ZAP to convert the TCR into a ‘catalytic unit’ that amplifies antigenic stimulation, which is achieved by phosphorylating the cycle of recruitment, activation, and release of ZAP70 kinase on the TCR [42]. 

Despite the promising results, our study has several limitations that need to be addressed. Although the validity of the TCRRG-based signature has been confirmed across multiple independent HNSCC cohorts, further validation through large-scale independent studies remains essential for fully establishing the signature’s reliability and wide applicability. Additionally, this analysis did not include key clinical parameters such as therapeutic resistance and molecular features. The available TCGA HNSCC dataset, as well as the GSE41613 and GSE65858 datasets, provided limited or no information regarding these essential clinical and molecular attributes. Moreover, our case series lacks the immunohistochemical evaluation, and in particular, the Combined Positive Score (CPS). Combining CPS and our TCRRG-based signature might be important for the diagnosis and therapeutic guidance of patients with HNSCC [43]. Pathologists play a pivotal role in this process, as they are tasked with determining PD-L1 expression while navigating challenges such as the selection of appropriate specimens, the intrinsic heterogeneity of PD-L1 expression in tumors, and the impact of prior treatments [44]. Consequently, pathologists provide crucial diagnostic information that informs therapeutic decision-making. They are responsible for the careful selection of sample types for analysis, the use of appropriate diagnostic platforms and clones, and ensuring interobserver consistency [45]. Their expertise is vital for interpreting the complexities of PD-L1 expression and its evaluation, which is fundamental to the effective application of targeted therapies in oncology. Thus, further studies that integrate detailed clinical and molecular information are necessary to understand the clinical importance of these seven genes in the context of HNSCC.

## 5. Conclusions

In summary, we effectively constructed and verified a reliable prognostic signature utilizing TCRRG that exhibits a substantial correlation with the unfavorable clinical outcome among patients with HNSCC. This signature may function as a significant molecular biomarker for tracking the development of HNSCC and may provide critical insights into the selection of suitable therapeutic approaches.

## Figures and Tables

**Figure 1 cancers-15-05495-f001:**
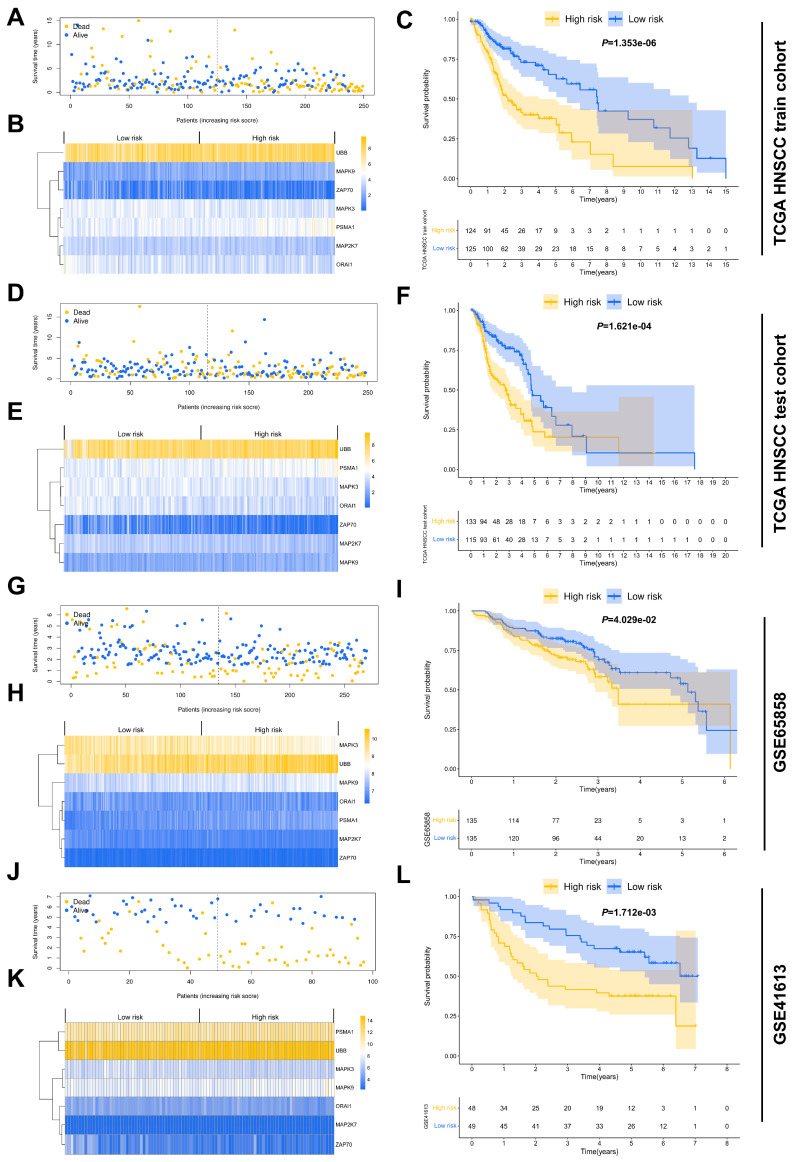
Development and validation of a TCRRG-based risk signature: (**A**) A scatter plot representing OS and survival status distribution in the low- and high-risk groups of the TCGA HNSCC train cohort. (**B**) A heatmap displaying differential expressions of specified TCRRGs between low- and high-risk groups in the TCGA HNSCC train cohort. (**C**) The survival difference between high- and low-risk groups in the TCGA HNSCC train cohort. (**D**–**L**) For the validation cohorts, the scatter plot depicts OS and survival status distribution in the low- and high-risk groups. Heatmaps illustrate differential expression levels of specified TCRRGs between these risk groups, while survival differences between high- and low-risk groups are further explored.

**Figure 2 cancers-15-05495-f002:**
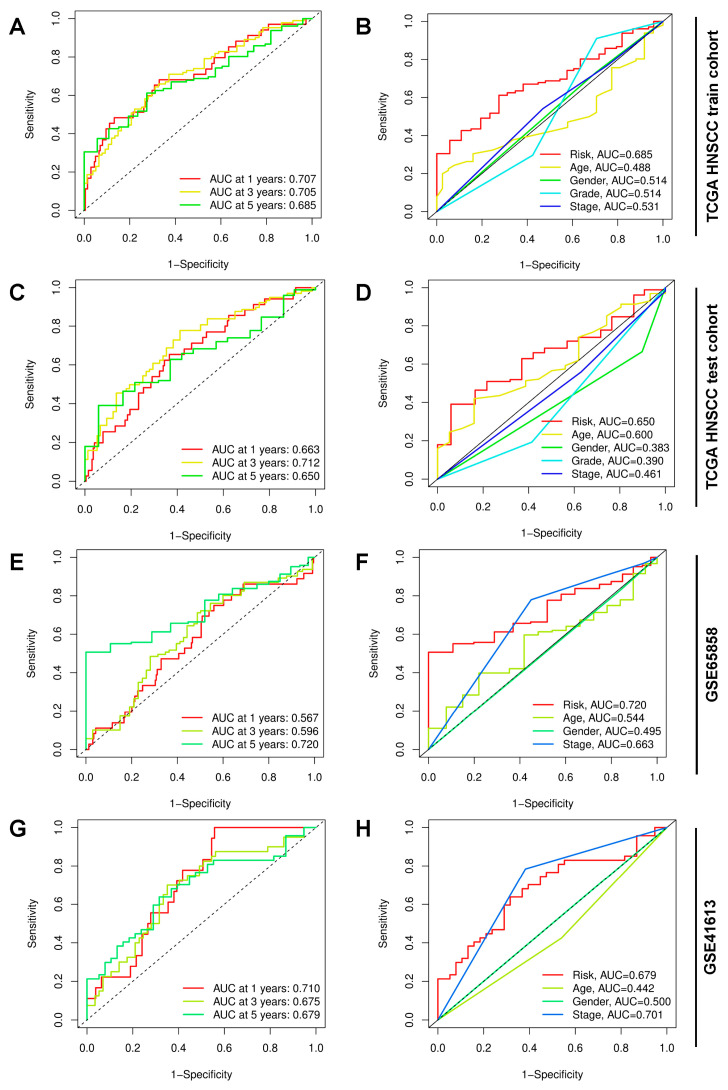
Evaluating the predictive precision of the TCRRG-based risk signature: (**A**) An ROC curve used to evaluate the prognostic signature’s predictive efficiency at specific time points in the TCGA HNSCC train cohort. (**B**) The predictive efficiency was compared between risk score and other clinicopathological parameters, including age, gender, grade, and stage, in the TCGA HNSCC train cohort. (**C**–**H**) In the indicated validation cohorts, ROC curves demonstrate the prognostic signature’s predictive efficiency at certain time points. Simultaneously, comparative analyses are conducted to assess the predictive efficiency of risk score against other clinicopathological parameters, including age, gender, grade, and stage.

**Figure 3 cancers-15-05495-f003:**
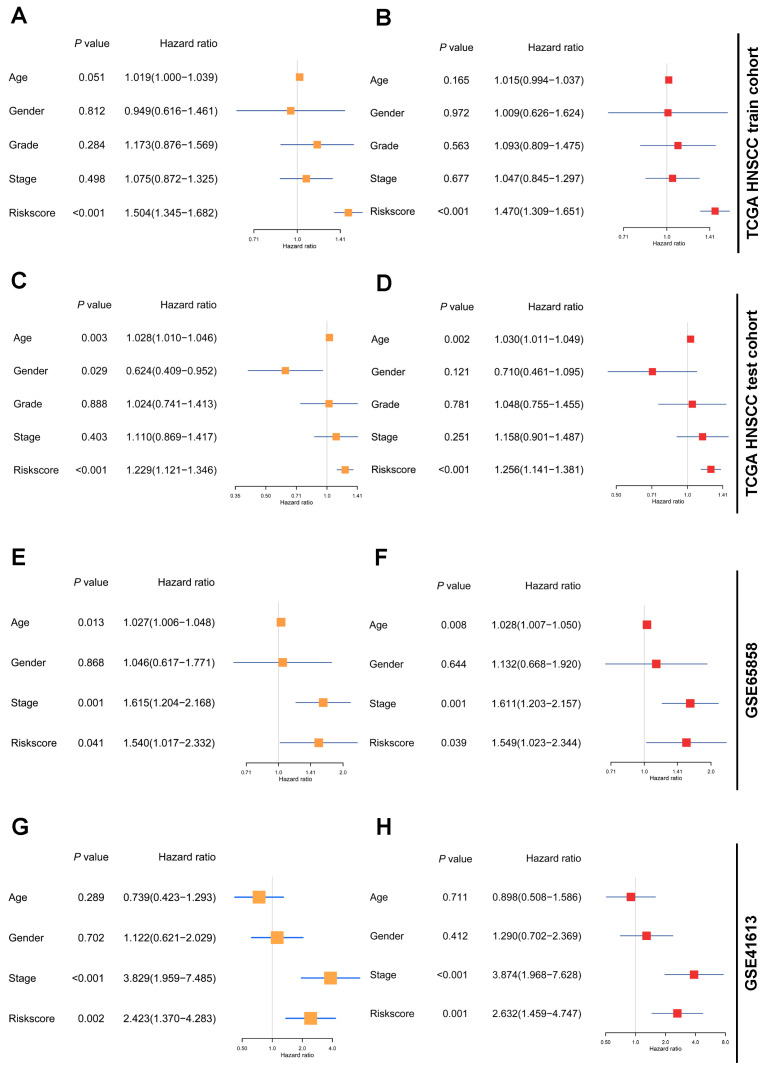
Independence of the TCRRG-based risk signature as a prognostic factor for HNSCC: (**A**) Univariate analysis of risk score in conjunction with clinicopathological parameters in the TCGA HNSCC train cohort to identify indicators significantly associated with OS. (**B**) Multivariate analysis of the risk score alongside clinicopathological parameters in the TCGA HNSCC train cohort to identify independent prognostic factors. (**C**–**H**) Univariate and multivariate analyses of risk score and clinicopathological parameters in the indicated validation cohorts for OS correlations and independent prognostic factor determination. Yellow squares represent univariate analysis, and red squares represent multivariate analysis.

**Figure 4 cancers-15-05495-f004:**
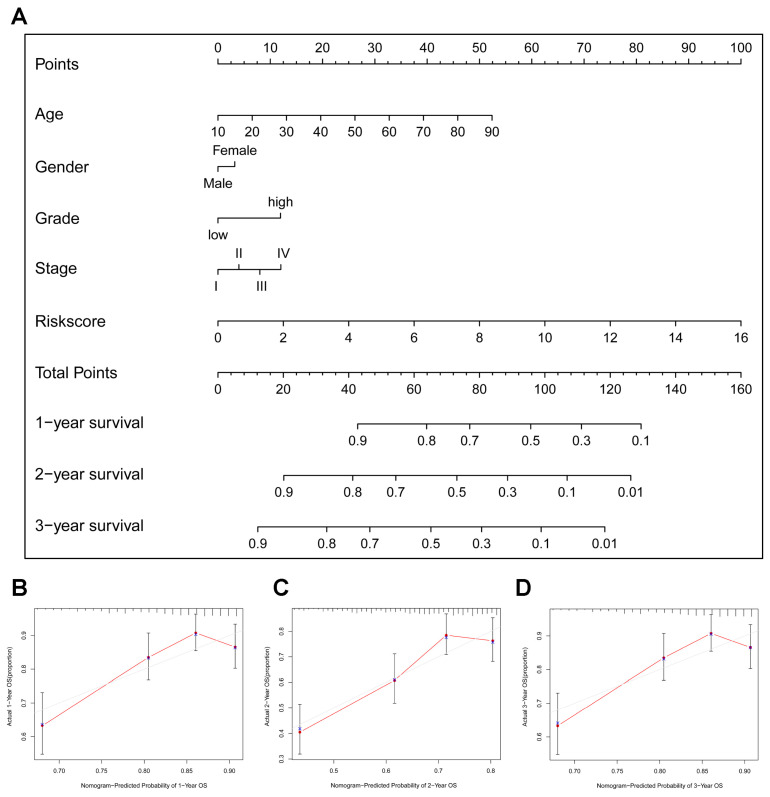
Construction and validation of a nomogram model integrating risk score and clinicopathological parameters: (**A**) A nomogram predictive model was built using the risk score and various clinicopathological parameters. (**B**–**D**) Internal validation of the nomogram’s predictive accuracy was conducted via calibration plots for 1-year (**B**), 2-year (**C**), and 3-year (**D**) OS predictions. The gray line represents the predictive curve, and the red line represents the actual curve.

**Figure 5 cancers-15-05495-f005:**
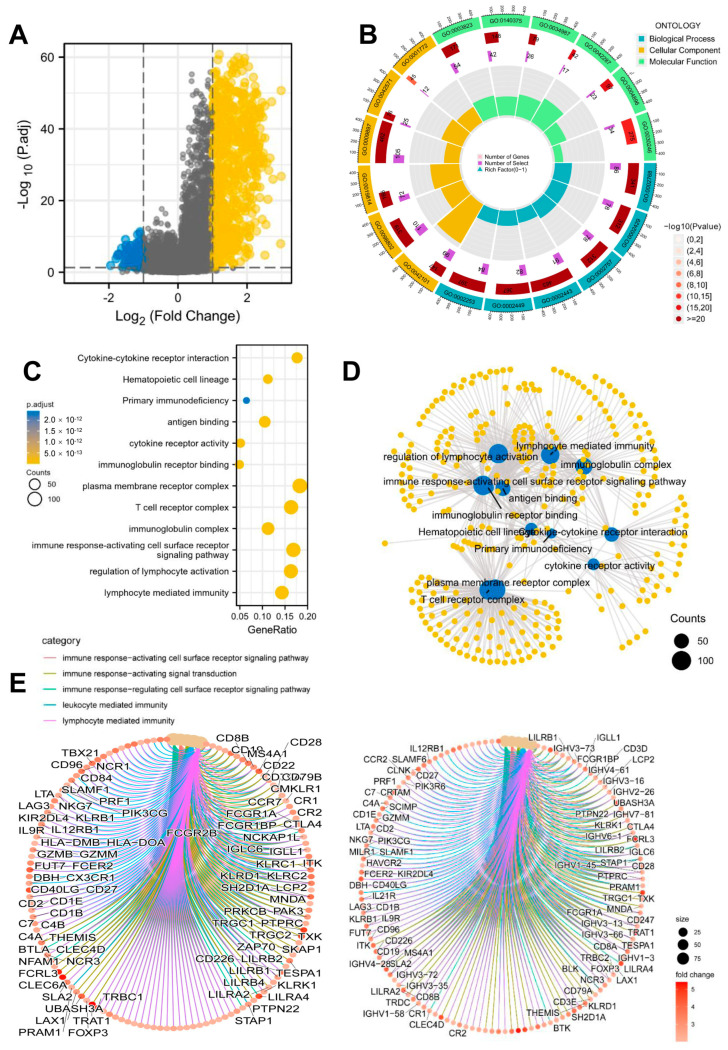
Functional enrichment analysis of DEGs between high- and low-risk groups: (**A**) The volcano plot illustrated significantly upregulated and downregulated genes in the high-risk and low-risk groups within the TCGA HNSCC cohort. (**B**) GO analysis of DEGs. (**C**) KEGG analysis of DEGs. (**D**) Visualization of the interconnected network resulting from GO and KEGG analyses. (**E**) Both GO and KEGG analyses highlighted enrichment in immune-related biological processes.

**Figure 6 cancers-15-05495-f006:**
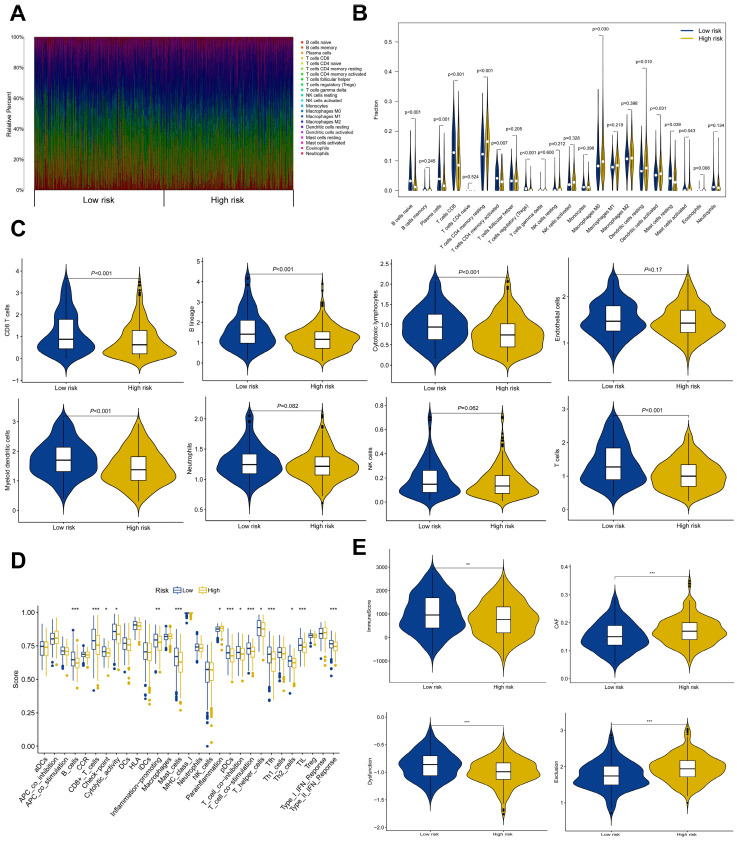
Differential immune characteristics between low- and high-risk groups in the TCGA HNSCC cohort: (**A**) Bar plot displaying the proportions of 22 unique types of immune cells in the low- and high-risk groups. (**B**) Violin plot showing the differences in immune infiltration status between low- and high-risk groups. (**C**) Violin plot illustrating the distinct distribution of immune cell types between low- and high-risk groups. (**D**) Box plot examining immune functions using TIDE scores in the low- and high-risk groups. (**E**) Comparison of immune score, CAF score, T-cell exclusion score, and T-cell dysfunction score between low- and high-risk groups. * *p* < 0.05. ** *p* < 0.01. *** *p* < 0.001.

**Figure 7 cancers-15-05495-f007:**
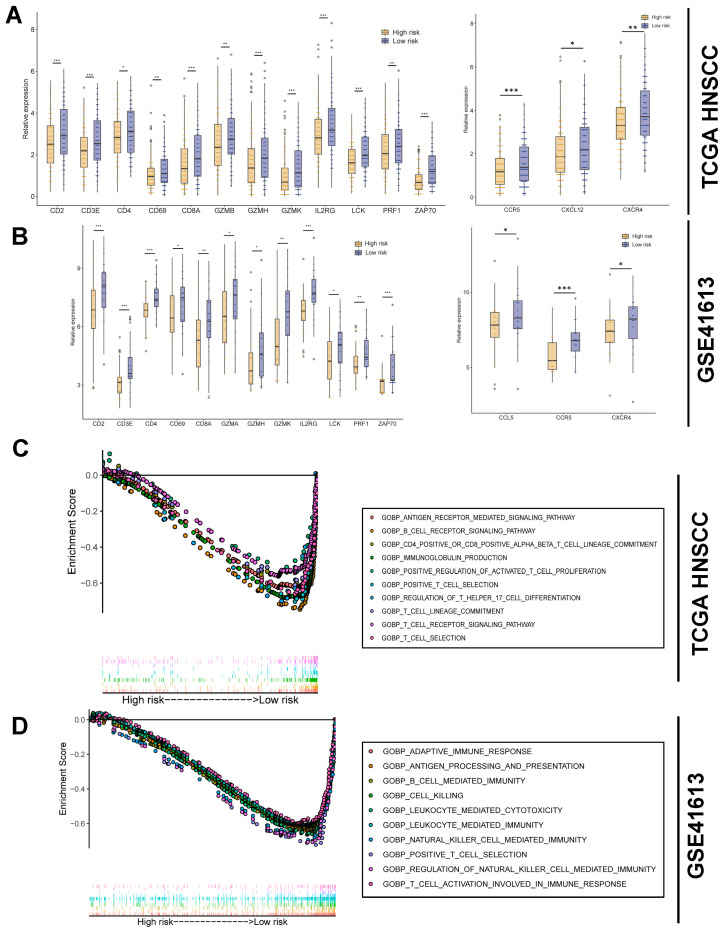
Risk score is associated with T-cell functions: (**A**) The differential expression levels of T-cell functionality and chemokine genes between the two risk groups in the TCGA HNSCC cohort. (**B**) The differential expression levels of T-cell functionality and chemokine genes between the two risk groups in the GSE41613 cohort. (**C**) GSEA analysis shows significant enrichment of genes in T-cell-related pathways in the TCGA HNSCC cohort. (**D**) GSEA analysis shows significant enrichment of genes in T-cell-related pathways in the GSE41613 cohort. * *p* < 0.05. ** *p* < 0.01. *** *p* < 0.001.

**Table 1 cancers-15-05495-t001:** Univariate analysis of TCR-related genes impacting the OS rate in HNSCC.

Gene	HR	HR.95L	HR.95H	*p* Value
*CSF2*	1.148047	1.01135	1.30322	0.032806
*INPP5D*	0.77208	0.596419	0.999477	0.049537
*MAP2K1*	1.626797	1.074057	2.463993	0.021605
*MAP2K7*	0.511056	0.319551	0.81733	0.00508
*MAPK3*	0.600922	0.383093	0.942607	0.026603
*MAPK9*	1.698728	1.06908	2.699215	0.024917
*ORAI1*	0.62139	0.440776	0.876014	0.006619
*PIK3R3*	0.763542	0.598236	0.974525	0.030215
*PSMA1*	1.675832	1.143652	2.455654	0.008085
*PSMA7*	1.657035	1.104644	2.485658	0.014647
*PSMD10*	1.566579	1.012122	2.424776	0.044007
*PSMD2*	1.478009	1.037221	2.106119	0.030602
*PSMD7*	1.741479	1.083136	2.799971	0.022046
*SKP1*	2.035235	1.27314	3.253516	0.002989
*UBB*	1.360397	1.011212	1.830161	0.041989
*UBE2D2*	2.191311	1.202443	3.99341	0.010406
*ZAP70*	0.731255	0.552287	0.968217	0.028852

**Table 2 cancers-15-05495-t002:** Multivariate analysis of TCR-related genes impacting the OS in HNSCC.

Gene	Coefficient
*MAP2K7*	−0.72216
*MAPK3*	−0.484
*MAPK9*	0.58118
*ORAI1*	−0.38196
*PSMA1*	0.477007
*UBB*	0.48365
*ZAP70*	−0.46462

## Data Availability

All data are included in the article and are based on publicly available databases.

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
