# Peer review of "Constructing a T-Cell Receptor-Related Gene Signature for Prognostic Stratification and Therapeutic Guidance in Head and Neck Squamous Cell Carcinoma"

_cancers, 2023, doi:10.3390/cancers15235495_

Round 1
Reviewer 1 Report
Comments and Suggestions for Authors
Authors aim was to construct a predictive signature derived from T-cell receptor-related genes to forecast the clinical outcomes in HNSCC. Utilizing consensus clustering analysis, we identified two distinct HNSCC clusters according to TCRRG expression. A TCRRG-based signature was subsequently developed and validated across diverse independent HNSCC cohorts. : The TCGA HNSCC cohort was divided into two clusters, demonstrating significant differences in overall survival and immune cell infiltration. The high-risk group was characterized by a suppressive immune microenvironment, in contrast to the low-risk group. The paper is well written and logically organized.
Minors:
- discuss the role of pathologist in providing diagnostic informations to guide decision making with particular reference to samples type, diagnostic platforms/clones and interobserver concordance among professionals. You should quote PMID: 35887569, PMID: 34530257, PMID: 34157159 because your body of reference has not considered this important, new literature evidence
- can you provide results regarding the CPS (combined positive score) among your case series?
- any insight related to the site of origin in Head & Neck region?
Comments on the Quality of English LanguageMinor editing of English language required
Author Response
Dear Reviewer,
We sincerely appreciate the time and effort you have dedicated to reviewing our manuscript, and your insightful feedback is truly valued. We have thoroughly examined your comments and made significant revisions to our manuscript in response. To assist you in evaluating the changes, we have submitted a detailed responses and the corresponding revisions in the re-submitted files.
Thank you very much again for your insightful comments. If any additional information is needed, please let me know.
Sincerely,
Xinyuan Zhao DDS, PhD

Reviewer 2 Report
Comments and Suggestions for Authors
This is an interesting study about constructing a T-cell receptor-related gene signature for prognostic stratification and therapeutic guidance in head and neck squamous cell carcinoma. The authors sourced gene expression profiles from The Cancer Genome Atlas (TCGA) HNSCC dataset, GSE41613 and GSE65858 datasets.
The paper is well written. However, some issues remain.
Please describe PD-1 and PD-L1 in the introduction when talking about immunotherapy.
The authors should specify if some patients included in the analyses underwent immunotherapy.
Moreover, a description of the treatments used for all the patients may be useful.
HPV status represents a prognostic factor for oropharyngeal cancer and should be added in the analyses.
Author Response
Dear Reviewer,
We express our profound gratitude for the dedication and effort you have extended in reviewing our manuscript. Your insightful comments and suggestions have been invaluable. In response, we have carefully reviewed and considered the suggestions. We have made revisions to our manuscript and included detailed, point-to-point responses along with the updated manuscript in the files we have resubmitted.
Thank you very much again for your insightful comments. If any additional information is needed, please let me know.
Sincerely,
Xinyuan Zhao DDS, PhD

Reviewer 3 Report
Comments and Suggestions for Authors
The authors aimed to show a TCRRG-based risk prediction model employing the TCGA HNSCC test cohort, GSE41613 and GSE65858. Additionally, a nomogram model based on the TCRRG risk signature was constructed to effectively predict the overall survival of HNSCC patients. Their results revealed significant increase in immune cell infiltration and an enhancement of immune functions in the low-risk group, compared to the high-risk group. These findings may provide insight into the underlying mechanisms responsible for the favorable clinical outcomes observed in the low-risk group.
Thus, I have no further comments against the manuscript.
Author Response
Dear Reviewer,
Thank you very much for summarizing our manuscript. Your comment that no further concerns about our manuscript is greatly appreciated.
Warm regards,
Xinyuan Zhao DDS, PhD

Round 2
Reviewer 2 Report
Comments and Suggestions for Authors
Thank you for improving the manuscript.